# Adiponectin Modulation by Genotype and Maternal Choline Supplementation in a Mouse Model of Down Syndrome and Alzheimer’s Disease

**DOI:** 10.3390/jcm10132994

**Published:** 2021-07-05

**Authors:** Melissa J. Alldred, Sang Han Lee, Stephen D. Ginsberg

**Affiliations:** 1Center for Dementia Research, Nathan Kline Institute, Orangeburg, NY 10962, USA; 2Departments of Psychiatry, New York University Grossman School of Medicine, New York, NY 10016, USA; 3Center for Biomedical Imaging and Neuromodulation, Nathan Kline Institute, Orangeburg, NY 10962, USA; SangHan.Lee@nki.rfmh.org; 4Child & Adolescent Psychiatry, New York University Grossman School of Medicine, New York, NY 10016, USA; 5Neuroscience & Physiology, New York University Grossman School of Medicine, New York, NY 10016, USA; 6NYU Neuroscience Institute, New York University Grossman School of Medicine, New York, NY 10016, USA

**Keywords:** adiponectin, type II diabetes mellitus, Down syndrome, Alzheimer’s disease, trisomy, selective vulnerability

## Abstract

Down syndrome (DS) is a genetic disorder caused by the triplication of human chromosome 21, which results in neurological and physiological pathologies. These deficits increase during aging and are exacerbated by cognitive decline and increase of Alzheimer’s disease (AD) neuropathology. A nontoxic, noninvasive treatment, maternal choline supplementation (MCS) attenuates cognitive decline in mouse models of DS and AD. To evaluate potential underlying mechanisms, laser capture microdissection of individual neuronal populations of MCS offspring was performed, followed by RNA sequencing and bioinformatic inquiry. Results at ~6 months of age (MO) revealed DS mice (the well-established Ts65Dn model) have significant dysregulation of select genes within the Type 2 Diabetes Mellitus (T2DM) signaling pathway relative to normal disomic (2N) littermates. Accordingly, we interrogated key T2DM protein hormones by ELISA assay in addition to gene and encoded protein levels in the brain. We found dysregulation of adiponectin (APN) protein levels in the frontal cortex of ~6 MO trisomic mice, which was attenuated by MCS. APN receptors also displayed expression level changes in response to MCS. APN is a potential biomarker for AD pathology and may be relevant in DS. We posit that changes in APN signaling may be an early marker of cognitive decline and neurodegeneration.

## 1. Introduction

Triplication of human chromosome 21 (HSA21) in utero causes Down Syndrome (DS), the primary genetic cause of intellectual disability (ID). DS occurs in an estimated 1 in 700 live births, and individuals with DS have multiple systemic deficits, including heart conditions, increased likelihood of leukemia and epilepsy, premature aging, and neurological deficits [1,2,3,4,5]. In recent decades, the lifespan of these individuals has significantly increased, although their healthspan still lags behind [1,4,6,7]. ID in individuals with DS is thought to be due to neurodevelopmental alterations in utero, including gray matter volume reductions, decreased neuronal cell densities, disorganization of laminar structures, astrocyte proliferation and hypertrophy [8]. DS pathology results in dysfunction in motor skills, hippocampal-dependent learning and memory, attentional function, language and communication [9], and cumulates in early onset of Alzheimer’s disease (AD) neuropathology by the mid-third decade of life [3,10,11,12,13,14,15,16]. The majority of adults with DS exhibit AD-like pathology, including neurofibrillary tangles, senile plaques, synaptic dysfunction and basal forebrain cholinergic neuron (BFCN) degeneration [3,10,11,12,13,14,15,16]. To facilitate mechanistic understanding and treatment development, several mouse models of DS and AD have been generated, the most well studied of which is the Ts65Dn (Ts) mouse model [17]. The Ts model recapitulates many of the neurological deficits seen in DS and AD, including hippocampal-dependent learning and memory deficits, BFCN degeneration and septo-hippocampal circuit dysfunction, notably CA1 pyramidal neuron and choline acetyltransferase (ChAT) activity deficits [18,19,20,21,22,23].

In addition to neurological deficits in aging individuals with DS, a higher incidence of metabolic syndrome/insulin resistance (MetS/IR) [24,25] and diabetes mellitus (DM) is observed in both children and adults [26]. A significant proportion of individuals with DS are classified as having adult onset or Type II Diabetes Mellitus (T2DM) [27,28]. MetS/IR as well as T2DM are recognized as risk factors for dementia associated with AD [29]. The protein hormones insulin, leptin and adiponectin (APN) are linked to MetS/IR, T2DM and the associated increased risk for AD pathology [30,31,32,33]. Studies in persons with DS have also shown dysregulation of these protein hormones [24,34]. Ts65Dn mice display MetS/IR pathology in peripheral studies, including increased glucose and decreased insulin levels [35,36]. However, similar studies focusing on the brain have not been performed to date. In brain, insulin is involved in neuronal growth, repair and signaling, and AD subjects display MetS/IR, often in the absence of frank T2DM [37]. In brain, leptin acts on neurons in the hypothalamus, hippocampus and brainstem to equilibrate glucose and lipid metabolism [38]. Rodent studies have shown in the aging brain reduction in the leptin response [39]. Both in vivo and in vitro application of leptin has beneficial effects in neurodegeneration models [40,41,42,43]. APN conveys neuroprotective effects in the brain, including protecting against ischemic brain injury and excitotoxicity [44,45,46], regulation of neurogenesis and proliferation in the hippocampus [47]. APN deficiency using APN^−/+^ and APN^−/−^ mice leads to reduced dendritic growth and spine density of granule cells in the dentate gyrus of the hippocampus [48]. Knockdown of APN in the brains of aged mice resulted in AD-like pathology and memory impairments [49], indicating that APN may play a critical role in neurodegeneration associated with AD. Studies in the periphery report conflicting findings on APN levels in children [24,50], and adults with DS [51,52,53,54,55,56]. Waragai et al. 2016, linked lower cerebrospinal fluid APN levels in AD with poor cognitive performance [52]. No studies to date have examined APN levels in the brain from individuals with DS or within mouse models of DS. 

Treating individuals with DS, as well as ameliorating cognitive decline associated with AD, are major public health issues. The most widely used FDA approved therapeutics for DS and AD are acetylcholinesterase inhibitors, which modulate cholinergic neurotransmission [57]. However, these drugs are ineffective at preventing disease progression and only treat symptoms. Thus, other therapeutic approaches are indicated and there is a current unmet need. 

Choline is an essential nutrient which is critical for biosynthesis of the neurotransmitter acetylcholine, is a key substrate of the phosphatidylethanolamine N-methyltransferase (PEMT) pathway, and is the primary dietary source of methyl donors [9,58]. Choline has been shown in rodent models to be essential for fetal brain development, with high levels of choline required for proper neural tube closure and brain development, often depleting maternal stores [43]. A potential treatment modality, maternal choline supplementation (MCS), in which increased levels of choline is delivered to developing pups during gestation and lactation, improves cognitive behaviors in several rodent models [9,58,59]. MCS in the Ts mouse model of DS and AD provides benefits including improved spatial cognition and attentional function, protects BFCNs, and normalizes hippocampal neurogenesis in adult offspring [9,23,60,61,62,63,64,65]. MCS increases ChAT-immunoreactive fiber staining level intensity in the hippocampus prior to BFCN degeneration in trisomic mice [61]. MCS also protects the cholinergic system in an amyloid-beta precursor protein (APP) overexpression model of AD [66]. Several studies using rodent models of neurodevelopmental disorders have demonstrated behavioral and morphologic benefits of MCS [67,68,69,70,71,72], indicating that MCS is neuroprotective and improves behavioral outcomes in multiple disease models. Further, from a translational perspective, MCS administered during the 3rd trimester in normal human pregnancies positively impacts several behaviors in normal infants, including increased performance in attaining developmental milestones [73]. In human plasma samples, decreased choline and choline metabolites were seen in T2DM/obese IR patients compared to obese insulin sensitive individuals, indicating that higher choline and metabolites may confer decreased risk of MetS/IR [74].

We previously identified the T2DM pathway as dysregulated within BFCNs in the Ts mouse at ~6 months of age (MO) [75]. To investigate underlying mechanisms driving T2DM dysregulation in DS, we examined protein hormones and receptors within the T2DM pathway in Ts brain compared to normal disomic (2N) littermates. We postulate dysregulation in the T2DM pathway may be attenuated by MCS. Accordingly, we tested levels of APN, insulin and leptin in the frontal cortex (Fr Ctx) of Ts and 2N littermates. We also examined the effects of MCS on these key protein hormones and APN receptors.

## 2. Materials and Methods

### 2.1. Mice and Maternal Diet Protocol

Animal protocols were approved by the Nathan Kline Institute/NYU Grossman School of Medicine Institutional Animal Care and Use Committee (IACUC) in accordance with NIH guidelines. Breeder pairs (female Ts65Dn and male C57Bl/6J Eicher x C3H/HeSnJ F1 mice) were purchased from Jackson Laboratories (Bar Harbor, ME, USA) and mated at the Nathan Kline Institute. Breeder pairs were assigned to receive one of two choline-controlled experimental diets: (*i*) control rodent diet containing 1.1 g/kg choline chloride (AIN-76A; Dyets Inc., Bethlehem, PA, USA), or (*ii*) choline-supplemented diet containing 5.0 g/kg choline chloride (AIN-76A; Dyets Inc.), as described previously [23,62,63]. The choline-supplemented diet provides approximately 4.5 times the concentration of choline consumed by the controls and is within the normal physiological range [76]. The control diet supplies an adequate level of choline, so the offspring are not choline deficient. Mice were given *ad libitum* food and water access [77,78]. Mice were maintained on a 12-h light-dark cycle under temperature- and humidity-controlled conditions. Pups born to choline supplemented (Ts+ and 2N+) or unsupplemented maternal choline (Ts and 2N) dams were weaned on postnatal day 21 and provided *ad libitum* access to water and the control diet. MCS and unsupplemented pups were housed upon weaning in standard cages (*n* = 2–4 mice per cage) containing paper bedding and several objects for enrichment (e.g., plastic igloo, t-tube and cotton square). Tail clips were taken and genotyped [79] at weaning and mice were aged to ~6 MO.

### 2.2. Tissue Preparation

At ~6 MO, mice were sacrificed for brain tissue accession. Mice were given an overdose of ketamine and xylazine and perfused transcardially with ice-cold 0.15 M phosphate buffer to clear blood and other potentially confounding peripheral markers out of the brain [77,78,80,81]. Brain tissues were accessed from unsupplemented normal choline Ts65Dn offspring (Ts; *n* = 10), MCS Ts65Dn offspring (Ts+; *n* = 10), unsupplemented normal choline disomic offspring (2N; *n* = 10) and MCS disomic offspring (2N+; *n* = 10) male mice with littermates between 2N and Ts mice used when possible (age range: 5.8–6.4 MO, mean age 6.0 MO). In order to obtain enough tissue sample to perform assays, left Fr Ctx tissue from each mouse brain was dissected using standard coordinates from the mouse brain atlas [82]. Dissected brain tissues were either flash frozen or kept on wet ice for homogenization immediately following brain accrual. Tissue was homogenized using ice-cold Tris homogenization buffer (20 mM Tris-Cl (pH 7.4), 1 mM EGTA, 1 mM EDTA and 0.25 M sucrose) with a protease inhibitor cocktail (1:1000, I3786, Sigma-Aldrich, St. Louis, MO, USA and 1 mM PMSF, ThermoFisher, Waltham, MA, USA) using 1.5 mm zirconium beads on Beadbug homogenizer (Benchmark Scientific, Sayreville, NJ, USA) for 30 s at 4000 rpm. Post homogenization, samples were kept on ice and cell debris was spun down at 2500× *g* for 5 min at 4 °C. Supernatant was aliquoted to fresh tubes for isolation of mRNA, which was performed immediately following homogenization, or for protein assays. Each protein assay was performed using tissue homogenate aliquots stored at −80 °C. RNase-free precautions were employed, and solutions were made with 18.2 mega Ohm RNase-free water (Nanopure Diamond, Barnstead, Dubuque, IA, USA). All consumables were certified RNA/DNA, nuclease and endotoxin free.

### 2.3. RNA Purification

RNA from Fr Ctx was purified using the miRNeasy Mini kit (Qiagen, Hilden, Germany) according to manufacturers’ specifications. A DNase digestion to remove genomic DNA was performed twice sequentially before the final washes and RNA purification. RNA quality control was performed at a 1:5 dilution to preserve RNA for downstream applications (RNA 6000 Pico kit, Agilent, Santa Clara, CA, USA).

### 2.4. RT-qPCR

cDNA was synthesized from equal amounts of RNA in a 50 µL sample reaction using random hexamers, as described previously [77,78,80,81,83,84] (Ts, Ts+, 2N, 2N+; *n* = 10/condition). RT-qPCR was performed using 1 µL of cDNA and Taqman qPCR primers for APN receptor 1 (*Adipor1*, Mm01291334_mH) and APN receptor 2 (*Adipor2,* Mm01184032_m1; Life Technologies, Grand Island, NY, USA) in triplicate using a real-time PCR cycler (PikoReal, ThermoFisher), as previously described [77,78,80,81,83,85,86]. The ddCT method was employed to determine relative gene level differences between groups [86,87,88]. Glucuronidase Beta (*GusB,* Mm01197678_m1) and 45S pre-ribosomal RNA *(Rn45s;* Rn03928990_g1) PCR products were utilized as controls. *Rn45s* was selected as the control housekeeping gene for ddCT quantification. Negative controls consisted of the reaction mixture without input RNA. The four study groups (Ts, Ts+, 2N, and 2N+) were compared to identify significant genes along with significant pairs with respect to PCR product synthesis for *Adipor1* and *Adipor2*. For each gene, the PCR product synthesis was modeled as a function of mouse study group (2N, 2N+, Ts, and Ts+), using mixed effects models with random effects to account for the correlation between repeated assays on the same mouse [77,78,80,81,89]. Contrasts of pairwise comparison of interest (e.g., 2N-Ts, 2N-2N+, Ts-Ts+, 2N+-Ts. 2N-Ts+) were subsequently constructed based on the fitted mixed effects model and tested by *t*-test from the model. *p*-Values were controlled by the false discovery rate (FDR) [90] due to the multiple comparisons employed. Significance was judged at the level α = 0.05, two-sided.

### 2.5. Protein Assays

Protein expression analysis was performed using the WES system (Protein Simple, Santa Clara, CA, USA) [91]. Protein samples were diluted in THB buffer 1:100 (*w*/*v*), with 1x final concentration of fluorescent molecular weight marker (provided by the manufacturer) and heated to 95 °C for 5 min, then cooled to 4 °C before loading onto the WES system plate with a molecular weight ladder. All blocking reagents, chemiluminescent substrate, separation and stacking matrices (Protein Simple) were dispensed to designated wells. Primary antibodies for ADIPOR1 (MA5-32249; 1:20; ThermoFisher), ADIPOR2 (NBP1-28641; 1:20; Novus Biologicals, Centennial, CO, USA or 14361-1-AP; 1:20; ProteinTech, Rosemont, IL, USA), β-Tubulin III (β-TUB III; MAB1195; 1:50; R&D Systems, Minneapolis, MN, USA) and HRP conjugated secondary antibodies (donkey anti-rabbit (A16029) and donkey anti-goat (A16005); ThermoFisher, 1:50 and rabbit anti-mouse (DM-002); Protein Simple, ready to use) were dispensed to designated wells. Plates were spun for 5 min at 1000× *g* and loaded onto a WES unit, where separation electrophoresis and immunodetection steps are fully automated within the capillary system. Instrument default settings were used with the exception of the protein loading run time, which was increased from 25 to 35 min. Digital images were analyzed with Compass software (Protein Simple), utilizing dropped lines for peak analysis area calculation. Detected proteins were compared to control β-TUBIII levels and reported as normalized percentage of control. Each protein was performed in triplicate on separate plate runs. Statistical analysis was conducted on each protein normalized to β-TUBIII and modeled as a function of the mouse study group (2N, 2N+, Ts and Ts+; *n* = 10/genotype/diet), using mixed effects models with random effect to account for the correlation between repeated assays on the same mouse [77,78,80,81,89]. Similar to the RT-qPCR analysis, contrasts of pairwise comparison of interest (e.g., 2N-Ts, 2N-2N+, Ts-Ts+ and 2N+-Ts. 2N-Ts+) were generated using the fitted mixed effects model and tested by t-test from the model with *p*-values controlled by FDR [90]. Significance was judged at the level α = 0.05, two-sided.

ELISA assays were performed on Fr Ctx tissue homogenates (1:10 *w*/*v* in THB buffer) in duplicate for each sample according to the manufacturer’s specifications (Invitrogen) with the exceptions as follows. For the insulin ELISA (EMINS, ThermoFisher), assay standards were added at 3.125 µIU/mL and 1.5625 µIU/mL and samples were run at 1:2 dilution of brain homogenates (*w*/*v*). For the APN ELISA (KMP0041, ThermoFisher), an assay standard was added at 0.0625 ng/mL and samples were run at 1:4 dilution of brain homogenates (*w*/*v*), as recommended by manufacturer. The Leptin ELISA (KMC2281, ThermoFisher) assay was performed as recommended with 1:4 dilution of brain homogenates (*w*/*v*). Statistical analysis was conducted using mixed effects models, as described above [77,78,80,81,89,90]. Significance was judged at the level α = 0.05, two-sided.

## 3. Results

### 3.1. Protein Hormone Levels in Trisomic Mice and the Impact of MCS

We previously showed dysregulation within the T2DM pathway in male Ts mice compared to 2N littermates at ~6 MO [75]. Accordingly, we used Fr Ctx tissue, which receives a prominent cholinergic input from BFCNs to interrogate three protein hormones dysregulated in human MetS/IR and T2DM: APN, insulin and leptin [30,31,32,33] in trisomic mice, with the hypothesis that MCS would have a beneficial effect. APN levels were significantly downregulated by approximately two-fold in Ts Fr Ctx (mean 9.53 ng/mL; *p* < 0.001; Figure 1A) compared to 2N (mean 18.89 ng/mL). MCS had a beneficial effect on trisomic APN levels, as a significant increase was observed in MCS (Ts+) offspring compared to normal choline (Ts) offspring (*p* < 0.01, Figure 1A). Ts+ APN levels were significantly lower than 2N levels (independent of MCS treatment), indicating that MCS partially rescues APN deficits in this trisomic mouse model (Ts+ mean 14.11 ng/mL; 2N compared to Ts+ *p* < 0.05; 2N+ compared to Ts+ *p* < 0.01). MCS had no significant effect on 2N Fr Ctx APN levels (2N mean 18.89 ng/mL versus 2N+ mean 17.59 ng/mL). ELISA assays did not reveal significant differences by genotype or diet for insulin (Figure 1B) or leptin (Figure 1C) in Fr Ctx.

### 3.2. APN Receptor mRNA and Protein Levels in Trisomic Mice and the Impact of MCS

APN signals through its receptors *Adipor1* and *Adipor2* which are expressed within multiple regions of the brain, including Fr Ctx [92,93,94,95]. RT-qPCR analysis revealed no significant differences in *Adipor1* or *Adipor2* levels by genotype in Fr Ctx (Figure 2A). However, a diet effect was observed as a significant increase in *Adipor1* (31.8%; *p* < 0.001) and *Adipor2* (28.2%; *p* < 0.05) was found in 2N mice due to MCS (e.g., 2N+ versus 2N). PCR product levels were also significantly higher in 2N+ offspring compared to Ts+ offspring (*Adipor1*, 2N+-Ts+, 16.1%; *p* < 0.05; *Adipor2*, 2N+-Ts+, 37.7%; *p* < 0.05; Figure 2A). A significant increase was seen in *Adipor1* PCR product levels by diet in trisomic mice (e.g., Ts+ versus Ts; 23.3%; *p* < 0.05; Figure 2A), but not for *Adipor2* PCR product levels. In terms of Fr Ctx protein levels, no significant differences were observed in ADIPOR1 protein expression by genotype or diet (2N versus Ts; Figure 2B). A significant decrease in ADIPOR1 was found in trisomic choline supplemented offspring compared to normal choline 2N offspring (Ts+ versus 2N; 57.6%; *p* < 0.05; Figure 2B). ADIPOR2 protein expression levels showed no differences due to genotype (2N versus Ts) or by 2N diet (2N versus 2N+; Figure 2B). A significant decrease in ADIPOR2 expression was observed by diet in trisomic choline supplemented offspring compared to normal choline trisomic offspring (Ts+ versus Ts; 61.1%; *p* < 0.05; Figure 2B).

## 4. Discussion

We utilized the well-established Ts65Dn mouse model of DS and AD to interrogate potential changes in several key protein hormones in the T2DM pathway within Fr Ctx brain tissue in the context of MCS. Through bioinformatic inquiry, our previous expression profiling study revealed significant dysregulation throughout the T2DM pathway within BFCNs in ~6 MO Ts mice [75], which we postulated was due to changes in protein hormone levels. Prior studies have shown alterations in leptin, insulin and APN can have severe deleterious effects on synaptic plasticity, learning, and memory [37,96]. We demonstrated significant downregulation in Fr Ctx APN levels in trisomic mice which was partially rescued by MCS (Figure 1A). No parallel alterations in insulin (Figure 1B) or leptin (Figure 1C) were observed in Fr Ctx brain tissue either by genotype or diet, indicating that APN is selectively vulnerable in this system. Further, we showed MCS modulates *Adipor1* and *Adipor2* expression (Figure 2), suggesting that MCS may be a viable treatment for APN-related defects associated with DS and AD.

Downregulation of APN levels in trisomic mice is commensurate with findings in several human studies in DS [34,50], AD [52,56] and mild cognitive impairment with T2DM [97], where decreased APN concentrations are found. Studies in brain tissue from animal models of neurodegeneration show that APN and APN receptors impact signaling and neuronal function in the brain, including neuronal insulin resistance, synaptic plasticity, excitotoxicity, neurodegeneration and cognitive decline [48,49,96,98,99,100,101]. While leptin levels are increased in peripheral studies in persons with DS [24], no studies have reported changes in CNS. Further, insulin levels are decreased in peripheral studies [34], but when examined in human DS brain, there were no significant alterations [102], commensurate with our current findings. Importantly, the present study reveals deficits in APN levels are positively impacted by MCS. MCS is a well-tolerated, low cost easily administered nutrient therapeutic that can be delivered throughout the perinatal period [9,103,104], which has previously shown to have long-term beneficial effects on basocortical and septohippocampal dependent memory tasks [60,63,64,65]. The current results link increased APN levels with MCS, suggesting that early choline supplementation could potentially provide lifelong benefits for individuals with DS and T2DM through an increase in brain APN concentrations, potentially normalizing dysregulation seen in the T2DM pathway. Prior to this study, a question whether peripheral administration of therapeutics could positively impact brain level APN deficiencies was posed [96]. The present translational study in a well-established mouse model of DS and AD indicates that MCS can result in positive functional changes in brain APN levels, suggesting that this early dietary intervention is efficacious in terms of altering APN signaling in the brain.

While APN levels benefited from MCS administration in this model system, we also measured mRNA and protein levels of the APN receptors ADIPOR1 and ADIPOR2. In contrast to APN, no changes in ADIPOR1 or ADIPOR2 mRNA or protein levels were seen due to the genotype in Fr Ctx at ~6 MO (Figure 2), which may reflect compensatory mechanisms seen in these mice in early adulthood, or regional/cellular specificity in gene expression changes. Previous work in an AD mouse model found no changes in *Adipor1*, but decreased *Adipor2* levels in Fr Ctx [100,101]. No significant changes in protein levels for either receptor was reported [100], consistent with our findings in the trisomic mouse model. Knockdown of *Adipor1* expression resulted in neurodegenerative phenotypes, spatial learning and memory deficits, and insulin signaling dysfunction in the hippocampus [99]. Further, delivery of an APN nonapeptide mimetic which activates APN receptors restores synaptic function in *Adipor1* knockdown mice [101]. In human brain tissue, *ADIPOR1* displays high expression in the nucleus basalis of Meynert (NBM) of the basal forebrain [93]. This finding is quite relevant, as cholinergic basal forebrain neurons of the NBM supply cholinergic innervation to the cortical mantle and are selectively vulnerable in DS and AD [3,16,105,106,107]. Therefore, loss of APN expression in trisomic mice may be linked to BFCN degeneration and could be a selective alteration found in basal forebrain [98] or in the septohippocampal and/or basocortical circuits, as evidenced by our current findings in Fr Ctx. The studies conducted by Várhelyi et al., 2017 [100] and Ali et al. 2021 [101] on *Adipor1* and *Adipor2* suggest that RNA and protein changes are not be expected by genotype and may be driven later in the DS mice by BFCN degeneration or loss of APN expression in the brain.

Increased choline through MCS delivery upregulated *Adipor1* levels, but not *Adipor2* levels, in trisomic mice (Figure 2A), which may be due to the differential expression levels and interactions of these specific APN receptors within the basocortical circuit. We posit these two APN receptors may have unique expression and signaling properties, as increases in *Adipor2* levels, but not *Adipor1* levels, were seen in a stress paradigm [100]. Since we found that *Adipor1* levels are MCS responsive, this may indicate that BFCN degeneration seen in trisomic mice by ~6 MO may pace or precede changes in Fr Ctx dysfunction including APN signaling.

A caveat of examining APN concentrations in brain is that there are major changes in the functional form and relative amounts dependent on the area tested [96,108], and functional forms were not tested in this paradigm. Measurements of APN and APN receptors in the basal forebrain and other relevant regions in trisomic mice across the lifespan are planned to examine the potential for age-related regional specificity and alterations in a functional form. APN levels here were only assessed in male mice in the present study. Since a study found increased APN plasma concentrations in healthy adults acted as a risk factor for dementia only in females [53], sex differences in APN expression levels must be considered in future studies, especially in the context of BFCNs in the Ts65Dn mouse [22]. Accordingly, an age matched female cohort is in development to evaluate APN and its cognate receptors. Further, studies involving the basocortical and septohippocampal circuit that evaluate potential benefits on behavioral outcomes are also planned. While MCS has shown significant efficacy in multiple mouse models of neurodegenerative disorders [9,69,109,110], additional studies are required in humans as both positive and negative outcomes of clinical trials have been reported [73,111,112,113,114]. Moreover, preclinical and clinical trials are required to determine if MCS will be beneficial in the context of MetS/IR and/or T2DM, including in the study of leptin-based and insulin-derived models of diabetes [115,116]. Additional treatments affecting APN and downstream activators include Os-pep, a novel nonapeptide APN mimetic, which binds to APN receptors [101] and activin, which may be a potential therapeutic target for reduction of amyloidogenic proteins during AD progression [117].

In conclusion, alterations in APN and APN receptors, which may be subtype specific to vulnerable brain regions and functional circuits involved in memory and executive function, namely the septohippocampal and basocortical systems, respectively, could represent novel targets for therapeutic intervention in DS and AD in the context of MetS/IR and T2DM. This initial study provides proof-of-concept for this contention based upon findings in a well-established trisomic model and the significant impact of MCS on APN and *Adipor1* levels. Further studies at the basic and translational levels are needed to examine the clinical applications of APN treatment in age-related neurodevelopmental and neurodegenerative disorders.

## Figures and Tables

**Figure 1 jcm-10-02994-f001:**
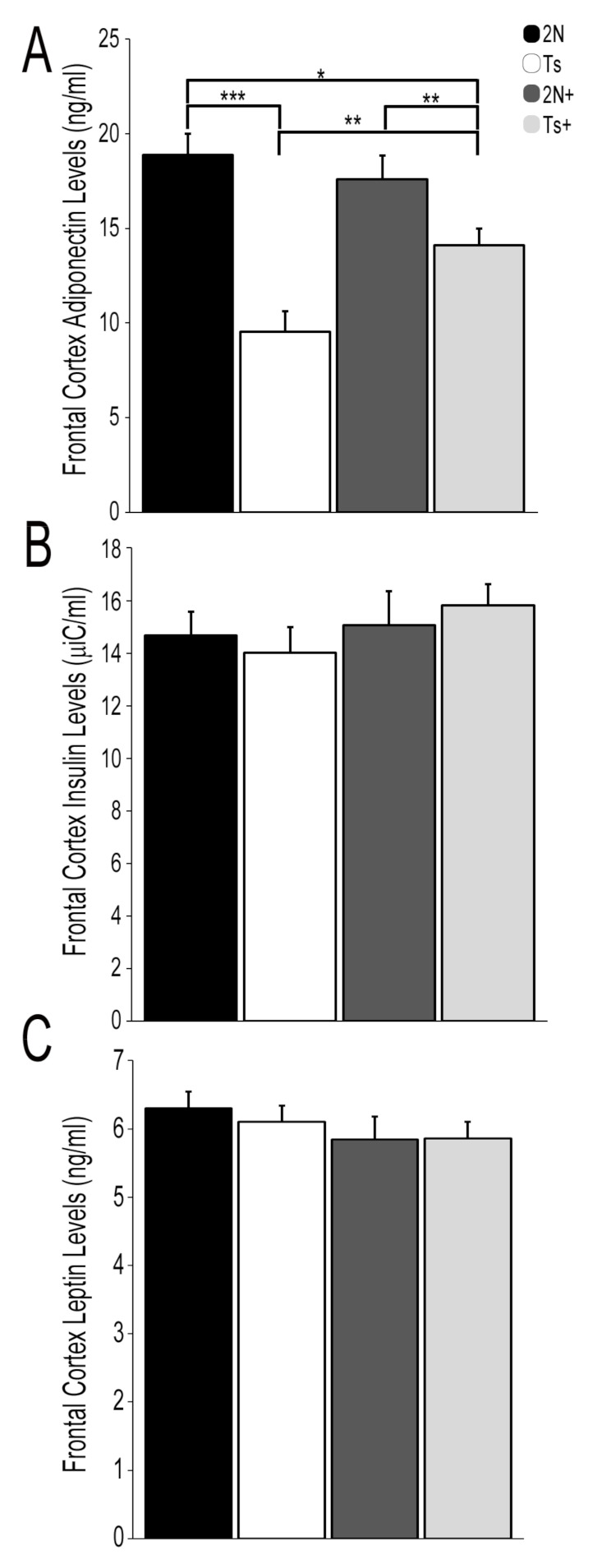
Protein hormone levels for select T2DM pathway members in the Fr Ctx of Ts65Dn mice and 2N littermates in relation to MCS. (**A**) APN levels were significantly downregulated in Ts mice compared to 2N littermates. Partial rescue was found with MCS in 2N+ and Ts+ mice, as APN levels were upregulated relative to offspring from a choline normal diet. (**B**) Insulin levels were not differentially regulated by genotype or diet in Fr Ctx. (**C**) Leptin levels were not differentially regulated by genotype or diet in Fr Ctx (Error bars are standard error of mean (SEM); * *p* < 0.05, ** *p* < 0.01, *** *p* < 0.001).

**Figure 2 jcm-10-02994-f002:**
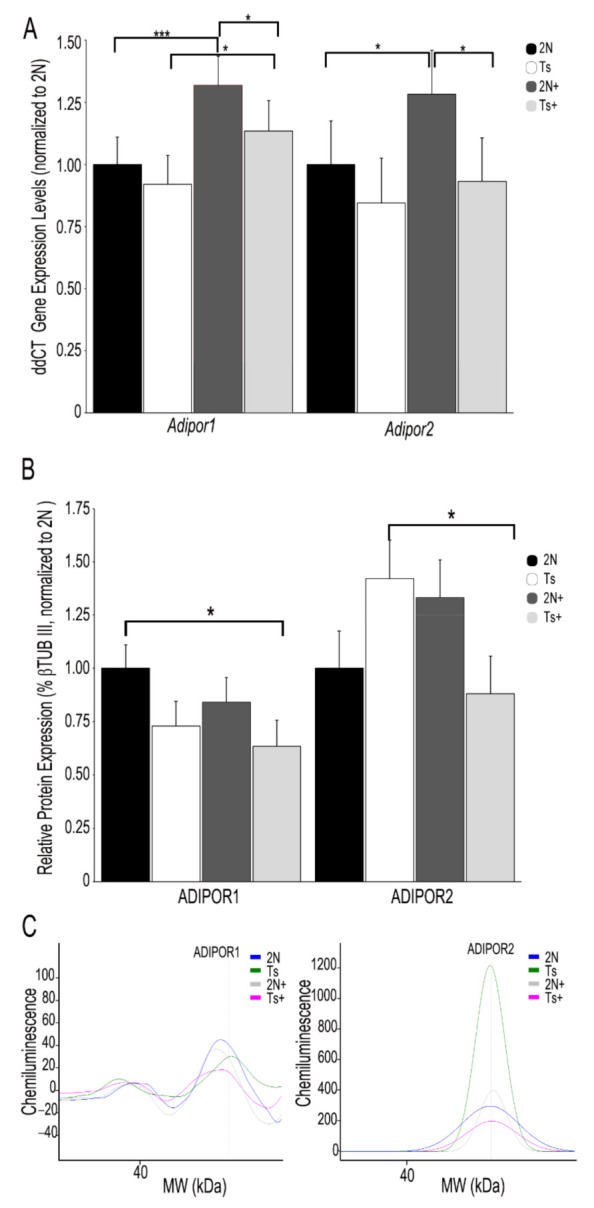
Gene and encoded protein levels of APN receptors ADIPOR1 and ADIPOR2 in the Fr Ctx of Ts65Dn mice and 2N littermates in relation to MCS. (**A**) *Adipor1* and *Adipor2* relative PCR product expression (normalized to 2N; ddCT with *Rn45s* as housekeeping gene) revealed no significant differences by genotype. A significant increase in *Adipor1* and *Adipor2* was found in 2N mice due to MCS. A significant increase in *Adipor1* was also found in Ts mice due to MCS. PCR product levels were significantly higher in 2N+ offspring compared to Ts+ offspring. (**B**) A significant decrease in ADIPOR1 was found in Ts+ offspring compared to 2N untreated animals. A significant decrease in ADIPOR2 was found in Ts mice due to MCS. (**C**) Representative digital signatures of each assayed protein raw expression levels (y-axis) and molecular weight (x-axis) from 2N, Ts, 2N+ and Ts+ Fr Ctx tissue homogenates. Key: Error bars = SEM; * *p* < 0.05, *** *p* < 0.001.

## Data Availability

Data analyzed within this study are included in this body of the manuscript. Data are also available from the corresponding author upon request.

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
