# Peer review of "Adiponectin Modulation by Genotype and Maternal Choline Supplementation in a Mouse Model of Down Syndrome and Alzheimer’s Disease"

_jcm, 2021, doi:10.3390/jcm10132994_

Round 1
Reviewer 1 Report
In this paper Dr. Alldred and colleagues has administered to a mouse model of Down syndrome (the Ts65Dn mouse) maternal choline supplementation (MCS) and they evaluated the effects of treatments on the offspring when they reach 6 months of age. MCS is a well-established treatment for Ts65Dn and it has been studied on several papers, every time with good results.
In the current study, the authors evaluated the effects of MCS on some components of type 2 diabetes mellitus (T2DM) pathway in the cortex of 6 MO male Ts65Dn mice. T2DM is one of the modern plague that affects a many people with Down syndrome and in the general population. In addition, it correlates with Alzheimer disease.
The idea at the base of this paper is certainly interesting, but it needs some major and minor revisions in order to be completed and re-evaluated. If comments below are addressed, I guess this article will be very helpful for scientific community working on Down syndrome.
Introduction
- “APN deficiency leads to reduced dendritic growth and spine density of granule cells in the dentate gyrus of the hippocampus” --> in which model?
- “Metabolic syndrome/insulin resistance (MetS/IR)” --> what about this condition in Ts65Dn mice?
- Can the authors highlight the rationale of MCS influence on T2DM pathway?
- “We previously identified the T2DM pathway as dysregulated within BFCNs in the Ts mouse at ~6 months of age (MO)” --> why didn’t the authors analyzed effect of MCD on T2DM pathway on BFCNs?
Methods
- How many mice per cage? How can the authors discriminate among the effects of cage enrichment and MCS? What is the levels of brain choline chloride in pups at birth?
- There were changes in pups’ mortality after treatment?
- I have not understood the procedure of brain storage. Mice were perfused, and brains were fast frozen or kept in dry ice? Why the authors perfused the animals if brains were frozen?
- Were consumables used in this study certified RNA and RNase free?
- Which kind of statistical analysis was performed for qRT-PCR? Was that a two-way ANOVA (with genotype and treatment as variables)? Followed by which post hoc test? I ask the author to add a paragraph in which they explain the statistic used. If there are no effects of variables, there’s no need to do a post hoc In those cases in which there were no effects of neither genotype nor treatment, why the authors have reported p values?
- For other experiments, did they use a post hoc test?
Results
- Long lasting effects are astonishing on APN levels. How the authors have interpreted these results? Why APN levels are still high after 6 month from treatment cessation?
- “APN deficiency leads to reduced dendritic growth and spine density of granule cells in the dentate gyrus of the hippocampus” (row 67) à Why the authors have investigated only CRTX?
- “including hippocampal-dependent learning and memory deficits, BFCN degeneration, and septohippocampal circuit dysfunction, notably CA1 pyramidal neuron and choline acetyltransferase (ChAT) activity deficits” (rows 51-53) --> it would be good to investigate APN, leptin and Insulin levels also in these brain regions. What about granule cell number of the DG? And connectivity?
- Figure 2B. A significant decrease in ADIPOR1 was found in Ts+ offspring compared to 2N untreated animals. A significant decrease 240 in ADIPOR2 was found in Ts mice due to 241 MCS (* p<0.05, ** p< 0.01, ***p<0.001) --> How do the authors interpret data for ADIPOR 2? How the author correlates no differences in ADIPOR1 and 2 and differences in APN levels?
Discussion
- Downregulation of APN levels in trisomic mice is commensurate with findings in several human studies in DS (34, 48), AD (50, 54), and mild cognitive impairment with T2DM (90) where decreased APN concentrations are found. Studies in brain tissue from animal models of neurodegeneration show APN and APN receptors impact signaling and neuronal function in the brain, including neuronal insulin resistance, synaptic plasticity, excitotoxicity, neurodegeneration, and cognitive decline (46, 47, 89, 91-94) à it would be confirmed in this study the correlation between APN regulation in trisomic mice and effective amelioration in “neuronal function in the brain”. For instance, with a cortex related behavior, or analyzing APN in pups and do developmental milestones tests.
- I partially disagree on the fact that Ts65Dn is a well established AD mouse model (Row 268). Ts65Dn has some common AD traits, but it has also difference. In addition, the Ts65Dn mouse at 6 MO (the age in which experiments are performed), display just few AD sings. In order to establish whether MCS could be affect AD symptoms in TS65Dn correlating this with T2DM, it would be investigated at an older age. It would be better whether the authors could cite some papers that claim what they state.
- APN is downregulated, while ADIPOR1 and 2 aren’t. What kind of compensatory mechanism has the authors though taking place?
- The authors have planned to analyze APN in BFCN in future studies. I think it would be analyzed in the current one. Since the authors think that APN is correlated with possible degeneration of BFCNs, it would be addressed whether MCS has positive effect APN in BFCNs region.
- As I wrote in introduction comments, It would be interesting to establish whether Ts65DN has metabolic syndrome. It is not clear whether the authors are speculating or they have performed experiments on that.
- How do they interpret the fact that in Ts65Dn Cortex there is no downregulation of leptin and Insulin? Do they have planned to analyzed levels of these proteins in other brain regions?
Coclusion
In conclusion section, the authors state as follows
“This initial study provides proof-of-concept for this contention based upon findings in a well-established trisomic model and the significant impact of MCS on APN and Adipor1 levels. Further studies at the basic and translational levels are needed”
I agree with this sentence and this is the reason why the title seems to be a little misleading. I recommend changing the title with something less catchy but more close to the results reported in the study.
Reviewer 2 Report
In "Maternal Choline Supplementation Rescues Adiponectin Defects, a Marker of Metabolic Syndrome, in a Mouse Model of Down Syndrome and Alzheimer’s Disease" the authors present their work analysing the effects of maternal Choline supplementation on a mice model mimicking brain characteristics of Down Syndrome (DS) and Alzheimer Disease (AD). They analyse in the frontal cortex by ELISA the concentration of different protein hormones related to Type II Diabetes Mellitus, disease related to DS. Their results show that in the transgenic animals, there is a reduction on the levels of adiponectin, that is partly recovered with the treatment. Then they check the levels of mRNA and protein expression of the adiponectin receptors in the frontal cortex, finding different effects on each condition making it hard to interpret the results.
The authors have a lot of publications using this paradigm with this model, but my feeling with the paper is that the authors do not explain enough some of their choices, reasoning or the functional implications. Why did they chose to analyse the frontal cortex? Reference 87: "Rastegar S, Parimisetty A, Cassam Sulliman N, Narra SS, Weber S, Rastegar M, et al. Expression of adiponectin receptors in the brain of adult zebrafish and mouse: Links with neurogenic niches and brain repair. J Comp Neurol. 2019;527(14):2317-33. " does not show this region. Is their expression significantly higher than other areas? Why not the hypothalamus, implicated in sugar blood levels? What are the functional implications of the partial rescue of adiponectin levels with the treatment? Do the animals have better score in behavioural tests? Do they have restored sugar blood level? Or it is not affected since insulin and leptin are not affected by the genotype nor the treatment?
The paper is highly based on their previous finding that T2DM pathway is dysregulated in the mice model at 6 months of age. Unfortunately, this reference : "Alldred MJP, S.C.; Lee, S.H.; Heguy, A.; Roussos, P.; Ginsberg, S. D. . Profiling Basal Forebrain Cholinergic Neurons Reveals a Molecular Basis for Vulnerability Within the Ts65Dn Model of Down Syndrome and Alzheimer’s Disease. Molecular Neuorbiology. 2021." is not published today, so I cannot read it and understand it. I do not doubt that their are trying to publish it, and it could come out tomorrow, but it should not be a reference today since I cannot check it (also, there isa typo on Neurobiology).
Finally, what is their interpretation of figure 2? Is the treatment good? Bad? Of course it affects its regulation and probably there are compensatory pathways, what is the conclusion of the treatment on this model in the frontal cortex? Specially since we do not find the same effect on the gene and protein expression. Are they different regulatory pathways in the transcription and traduction? Or is it due to the clearance?
What is the relationship of T2DM and AD? How adiponectin (or its regulation) could be a therapeutic target for reduction of amyloidogenic proteins during AD progression? Does MCS an effect on DS? These changes on the aetiology that later could influence on the development or not of AD?
Some minor details that I think that the paper should improve.
What is represented on the figures? SEM? Are the error bars the standard error? Or variance or what?
Since the paper does not have too many figures, I think they should include example gels or images that show what was quantified.
Not everybody knows or understand ddCT or Os-pep.
This phrase on line 300:"While MCS has shown significant efficacy in multiple mouse models of neurodegenerative disorders..." should be backed with some references.
The units on Fig2B "Relative protein expression (% BetaTubIII)", since they are around 1, I understand it is not a percentage. I am not an expert on his field, but I would say as a ratio or related to.
The statistical analysis should be explained with further detail, not just: "using mixed effects models with random effect to account for the correlation between repeated assays on the same mouse 148 (73, 74, 76, 77, 85). Significance was judged at the level α=0.05, two-sided." Were both variables significant? And their interaction? What post-hoc test are they using to compare between the different conditions (each column in the graphs)?
Round 2
Reviewer 2 Report
The authors have addressed and answered the criticisms of their first version of the article. I think that they have made an effort to improve the article and its discussion, so I will not ask for a major revision, but I still have some small concerns about improving it.
I found weird the new tittle, since it is an animal model, of course de genotype should affect the question in study. I will not say that it is imperative to change it, but I think they should consider it and bring up a new version.
You have already answered in comment 16 about ddCT and Os-pep, but I think it will be clearer to include some small explanation in the text. Something like: "and downstream activators include Os-pep ( a osmotin-based small adiponectin-mimetic peptide) which binds to APN receptors (94101)".
Thank you for noting in your answers when you had already answered to referee 1, but I don't have access to his/her comments nor your answers. And thank you for writing down the lines, but since they send me a pdf version with the changes tracked, the number of lines there were not the same that you wrote down.
With that in mind, I think there is an error in line 134 since the phrase was re-written. I have "In order to obtain enough tissue sample to perform assays, left Fr Ctx tissue from each mouse brain was using standard coordinates from the mouse brain atlas". Missing dissected?
Similarly, in line 354 "A caveat of examining APN concentrations in brain is there are major changes in the functional form and relative amounts dependent on the area tested". Missing that?
Congratulations for getting published reference 75
Author Response
Reviewer #2: We thank the Reviewer for her/his insightful comments.
- “I found weird the new tittle, since it is an animal model, of course de genotype should affect the question in study.”
Response: We respectfully disagree, as we found that APN levels were affected by genotype and MCS treatment but not levels of the APN receptors, which were only partially affected by MCS treatment. This is the basis for our title modification that was requested to highlight that adiponectin is modulated by both genotype and MCS treatment.
- “… comment 16 about ddCT and Os-pep but I think it will be clearer to include some small explanation in the text”
Response: We have modified the text to include “Os-pep, a novel nonapeptide APN mimetic,”
- Error in line 134
Response: Thank you for noticing that error, we have modified it to now state “tissue from each mouse brain was dissected using standard coordinates”
- Line 354 missing word
Response: Thank you for noticing that error, we have modified it to now state “examining APN concentrations in brain is that there are major changes”